# Persistent Methicillin-Resistant *Staphylococcus aureus* Bacteremia: Host, Pathogen, and Treatment

**DOI:** 10.3390/antibiotics12030455

**Published:** 2023-02-24

**Authors:** Joshua B. Parsons, Annette C. Westgeest, Brian P. Conlon, Vance G. Fowler

**Affiliations:** 1Department of Medicine, Division of Infectious Disease, Duke University Medical Center, Durham, NC 27710, USA; 2Department of Microbiology and Immunology, University of North Carolina at Chapel Hill, Chapel Hill, NC 27599, USA; 3Department of Infectious Diseases, Leiden University Medical Center, 2333 ZA Leiden, The Netherlands; 4Duke Clinical Research Institute, Durham, NC 27710, USA

**Keywords:** *Staphylococcus aureus*, persistent bacteremia, MRSA

## Abstract

Methicillin-resistant *Staphylococcus aureus* (MRSA) is a devastating pathogen responsible for a variety of life-threatening infections. A distinctive characteristic of this pathogen is its ability to persist in the bloodstream for several days despite seemingly appropriate antibiotics. Persistent MRSA bacteremia is common and is associated with poor clinical outcomes. The etiology of persistent MRSA bacteremia is a result of the complex interplay between the host, the pathogen, and the antibiotic used to treat the infection. In this review, we explore the factors related to each component of the host–pathogen interaction and discuss the clinical relevance of each element. Next, we discuss the treatment options and diagnostic approaches for the management of persistent MRSA bacteremia.

## 1. Introduction

With almost 20,000 deaths attributed to *Staphylococcus aureus* bloodstream infections in the USA in 2017, *S. aureus* bacteremia (SAB) is one of the most frequent and severe bacterial infections [1]. Methicillin-resistant *Staphylococcus aureus* (MRSA) is the most common cause of infections due to multidrug-resistant bacteria in the United States [2]. Bacteremia due to MRSA has long been associated with higher mortality rates than its more susceptible counterpart [3]. Although most studies have shown higher mortality rates, MRSA bacteremia (MRSAB) has only a slightly higher adjusted mortality compared to methicillin-susceptible SAB [4]. More recent high-quality studies in the field suggest a limited odds ratio (OR) or relative risk (RR) increase in death of around 1.3–1.8 [4].

We have learned over the decades that mortality in patients with SAB can be decreased through standardized clinical management practices such as obligatory infectious diseases consultation, routine echocardiography and follow-up blood cultures, and appropriate antibiotics [5,6,7,8,9,10]. Despite these insights, ≈25% of patients with SAB will die within 3 months of diagnosis [4].

One of the unique and disturbing features of SAB is the tendency of the organism to persist in the bloodstream despite the presence of microbiologically appropriate antibiotics. The phenomenon of persistent bacteremia remains poorly understood, and we lack great tools to identify who is at risk for persistent SAB.

This paper reviews the basic science and clinical literature behind persistent MRSAB. We discuss the contribution from the host and the pathogen in the pathophysiology of SAB.

### Persistent MRSAB

Persistent SAB is the strongest predictor of complicated SAB [11]. Multiple observational studies have identified the stark difference in mortality in patients with persistent SAB compared to those whose bacteremia promptly resolves [12,13,14]. One recent cohort of 884 patients with SAB (approximately one-third with MRSAB) determined that increasing duration of positive *S. aureus* blood cultures was associated with increased rates of metastatic complications, length of stay, and 30-day mortality [12]. The investigators concluded that each additional day of bacteremia was associated with a relative risk of death of 1.16 [12]. Another multinational cohort of 1588 patients with SAB found that 90-day mortality almost doubled (22 to 39%) when the duration of bacteremia increased from 1 day to 2–4 days [14]. Both studies underlined the severe consequences of persistent SAB. The consequences relating to treatment and further diagnostic evaluation are discussed later in this review.

Both the definition and the frequency of persistent SAB have evolved over the past two decades [15]. In the early 2000s, Fowler et al. defined persistent bacteremia as ≥7 days of positive blood cultures [16] on the basis of the median duration bacteremia in patients with MRSA [17,18]. The reliable therapeutic options for MRSAB during that era were limited to vancomycin only. As a result, the designation of persistent MRSAB had little therapeutic consequence, as in most clinical cases, the vancomycin was simply continued. Since then, however, several new antibiotics with effectiveness against MRSA have been approved by the Food and Drug Administration (FDA). One antibiotic, daptomycin [19], has been approved specifically for MRSAB. In addition, other antibiotics such as the fifth-generation cephalosporin ceftaroline [20] are frequently used off-label for MRSAB. Given the ability to use alternate antibiotics and some data supporting combination antibiotic therapy for MRSAB (discussed in Section 4.2), more recent reports have suggested modifying the definition of persistent MRSAB to include patients with positive blood cultures for as few as 2 days [14].This shorter duration allows for a “check point” to consider alternate therapy and broader diagnostic evaluation [21].

## 2. Host Factors Associated with Persistent MRSAB

### 2.1. Clinical Risk Factors

Numerous observational studies have identified independent patient risk factors for the development of persistent SAB (Table 1) [22,23,24,25,26,27,28]. A recurring theme is the presence of retained intravascular devices or foreign bodies, which are independently associated with persistent SAB [15,22,24,25,26,28]. Similarly, metastatic infection (including endocarditis, bone and joint infection), chronic renal failure, cirrhosis, and diabetes are also associated with persistent SAB [22,23,25,26,28]. The largest study was a nested case–control study examining risk factors for persistent SAB, performed by Chong et al., who included 483 patients with persistent SAB and 212 patients with resolving SAB [22]. In addition to the previously described risk factors, multivariate analysis revealed community-onset bacteremia, methicillin resistance, central venous catheter (CVC)-related infection, and vancomycin trough of <15 mg/L as risk factors for persistent SAB [22].

The majority of these studies do not distinguish methicillin-susceptible *S. aureus* (MSSA) from MRSAB, often citing vancomycin use as a risk factor for persistence [23,26]. Yoon et al. limited their investigation to MRSA only, identifying retention of implanted devices and metastatic infection of at least two sites as predictors of persistent MRSAB [24].

While these studies represent an important component in the understanding of persistent SAB and MRSAB, it currently comes as little surprise that unresolved sources of infection are the most frequently reported clinical risk factors for persistence. However, clinical risk factors only partially explain which patients develop persistent SAB. 

### 2.2. Host Genetic Variation and SAB

Genetic risk factors for infection have been identified in a wide range of infectious diseases [29]. A landmark study performed in the 1980s determined children of adults who experienced premature death due to infection were more likely to experience death due to infection themselves, suggesting a heritable basis for their infection risk [30]. Rare primary immunodeficiency syndromes such as chronic granulomatous disease, hyper-IgE syndrome, and Chédiak–Higashi have been associated with increased susceptibility to *S. aureus* infection [31,32,33,34]. Few studies have examined the genetic risk factors for *S. aureus* bloodstream infections and even less focus on persistent MRSAB. A fascinating study by Oestergaard et al. was performed in 2016 by examining a database consisting of almost all parents and children born in Denmark between 1954 and 2016 (*n* = 8,951,393) [35]. On the basis of 18,626 reported cases of SAB and 34,774 first-degree relatives, the investigators found that first-degree relatives of patients hospitalized for SAB were more likely to experience an episode of SAB themselves (standardized incidence ratio (SIR) of 2.49; 95% confidence interval (CI) 1.95–3.19). The risk was particularly notable in siblings of patients with SAB (SIR, 5.01; 95% CI 3.30–7.62) compared to parents (SIR, 1.96; 95% CI 1.45–2.67). While these data provide compelling evidence for heritable risk factors for acquiring SAB, the specific genetic defect remains unknown.

Three genome-wide association studies (GWAS) have been performed to identify host genetic variability that can predispose to SAB. Two smaller studies by Nelson et al. (361 SAB cases and 699 controls) and Ye et al. (309 cases and 2925 controls) did not identify single-nucleotide polymorphisms (SNPs) with genome-wide significance for risk of acquiring or severity of SAB [36,37]. A third larger GWAS study of 4701 SAB cases and 45,344 matched controls identified two SNPs that achieved genome-wide significance for altered susceptibility to *S. aureus* infection in individuals of European ancestry (rs35079132: *p* = 3.8 × 10^−8^, and rs35079132 *p* = 3.8 × 10^−8^) [38]. These loci were located near the HLA-DRA and HLA-DRB1 genes in the HLA class II region. Using admixture mapping, that same genetic region of European origin was also identified in African Americans as associated with SAB at a genome-wide level of significance [39]. This discovery was the first of its kind in *S. aureus* research and built on the enlarging body of evidence linking HLA haplotypes to susceptibility and severity of bacterial infection [40,41,42,43,44,45]. 

### 2.3. Host Genetic Variation and Persistent MRSAB

Despite the advances in our understanding of genetic risk factors for SAB, none of these studies addressed which genetic variants protect or place patients at risk of persistent methicillin-susceptible or methicillin-resistant SAB. A breakthrough discovery was made by Mba Medie et al., who identified a key association between genetic variation in the *DNMT3A* gene and protection against persistent MRSAB [46]. This elegant study performed whole-exome sequencing (WES) on a cohort of 68 patients with persistent MRSAB (*n* = 34), defined as persistently positive blood cultures for ≥5 days, and resolving MRSAB (*n* = 34), defined as blood culture positivity for <5 days. These patients were matched by sex, age, race, presence of implanted devices, diabetes mellitus status, and hemodialysis status. The study revealed a specific polymorphism (g.25498283A > C) in the DNA methyltransferase 3A intronic region of *DNMT3A* that was associated with a reduced risk of persistent MRSAB. The variant was identified in 61.8% of the cohort with resolving bacteremia and just 8.8% of patients with persistent bacteremia (*p* = 7.8 × 10^−6^). Examination of the DNA methylation patterns between patients with and without the g.25498283A > C mutation revealed significantly higher levels of methylation in gene-regulatory CpG island regions in patients expressing the homozygous genotype. Cytokine analysis also revealed significantly lower levels of anti-inflammatory cytokine interleukin-10 (IL-10) in acute phase serum from patients with resolving MRSAB compared to persistent MRSAB (114 pg/mL in persistent bacteremia patients vs. 13.1 pg/mL in resolving bacteremia patients; *p* = 0.009). IL-10 levels were also found to be lower in the subset of patients with the g.25498283A > C polymorphism, regardless of whether the serum was from patients with persistent MRSAB or resolving MRSAB (A/C: 18.9 pg/mL vs. A/A: 68.9 pg/mL in patients with persistent MRSAB and A/C:8.7 pg/mL vs. A/A:14.95 pg/mL in patients with resolving MRSAB). The proposed mechanism for decreased susceptibility to persistent MRSAB is thought to revolve around suppression of IL-10 production via DNA-methyltransferase-3A-mediated DNA methylation (Figure 1). While the exact role of IL-10 in promoting persistent MRSAB is unclear, this finding was consistent with prior studies that also found an association between elevated IL-10 and mortality from SAB and persistent SAB [13,47]. IL-10 is an immunosuppressive cytokine and is known to prevent the activation of Th1 helper T cells and subsequently can increase survival of some intracellular bacteria [48]. It is known that IL-10 signaling can suppress proinflammatory macrophage and cytokine production, resulting in less reactive oxygen species (ROS) and reactive nitrogen species (RNS) known to play a crucial role in fighting *S. aureus* and other pathogens [48,49,50,51,52]. One can hypothesize that the reduced IL-10 production in patients with the g.25498283A > C polymorphism allows for a more robust pro-inflammatory response, which assists with efficient clearance of bacteria from the bloodstream. However, more research in this field is needed to further unravel the complex mechanism.

A 2020 follow-up study by Chang et al. examined the DNA methylation pattern in leukocytes from 142 patients with persistent MRSAB (blood culture positive >5 days; *n* = 70) and resolving MRSAB (blood culture positive <5 days; *n* = 72) [53]. This study used advanced sequencing techniques to quantify and localize differences in the DNA methylome. DNA extracted from persistent MRSAB patients’ leukocytes exhibited significantly lower levels of methylation localized to binding sites for two transcription factors involved in immune regulation: signal transducer/activator of transcription 1 (STAT1) and CCAAT enhancer binding protein-β (C/EBPβ) (Figure 2). In contrast, the profile of the resolving MRSAB patients’ methylome localized differences in the histone acetyltransferase p300 and glucocorticoid receptor binding site. The mechanistic basis for these changes is proposed by the authors. Firstly, C/EBPβ has a role in emergency granulopoiesis [54], and the abundance of immature granulocytes arising from activation of the C/EBPβ gene may impair the ability of the immune system to assimilate the circulating bacteria, promoting persistence. Second, activation of STAT1 is known to induce T-helper cell polarization into the Th1, which tips the see-saw balance away from Th17-mediated interleukin-1 (IL-1) and interleukin 17 (IL-17) production known to mediate neutrophil recruitment and activation critical for bactericidal activity. Third, in resolving persistent MRSAB patients, the hypomethylation in glucocorticoid receptor and associated co-factor p300 histone acetyltransferase promoter regions likely helps counter-regulate the life-threatening pro-inflammatory response that occurs during bloodstream infections [55].

### 2.4. Biomarkers for Persistent SAB

These studies represent a potential breakthrough in unraveling the astonishingly complex genomic and epigenetic distinctions between patients with persistent MRSAB and resolving MRSAB. The clearest application of this discovery is the potential to identify patients at risk for persistent MRSAB, which could lead to alterations of initial therapy, expediting of additional diagnostic evaluation, and the capacity to improve clinical outcomes. Concurrent work in identifying biomarkers in patients with persistent SAB and persistent MRSAB has identified a handful of possible candidates. Using a threshold of blood cultures positive for >5 days to define persistent SAB, Guimaraes et al. identified eight proteins correlating with persistent SAB, with interleukin 17A (IL-17A), IL-10, and soluble E-selectin levels, showing the most robust association [47]. A follow-up study by Cao et al. found levels of IL-17A, IL-10, or soluble E-selectin levels were able to individually identify patients at risk of microbiologic failure and persistent SAB [56]. These biomarkers were more predictive than clinical risk factors known to increase risk for persistence (age, steroid use, hemodialysis, non-removable infection foci, hospital vs. community onset, and MRSA vs. MSSA). Given the association of persistent SAB with mortality, it is unsurprising that elevated IL-17A and IL-10 levels were each associated with increased mortality in this study [13,56]. 

While these discoveries are exciting and show promise for future diagnostic options to stratify patient risk for persistence, the clinical utility at the present day is hampered by availability only in specific academic centers and reliance on external laboratories to perform the tests. Fast turnaround time will be the key to the real-world use of these tests to identify patients at risk of persistent SAB. This could allow for early detection of persistent SAB and subsequently altered therapeutic and diagnostic strategies that could potentially save lives.

## 3. Pathogen-Associated Risk Factors for Persistent *S. aureus* Bacteremia

To survive and replicate in the bloodstream, *S. aureus* must avoid a barrage of host defenses while attempting to adhere to and proliferate upon an endothelial surface of the vasculature. The establishment of endovascular infection is a complex process requiring coordinated expression of multiple adhesins, exotoxins, and exoenzymes at various stages of infection. Meanwhile, *S. aureus* must resist or avoid phagocytosis by neutrophils and the resulting oxidative and non-oxidative burst, in addition to the circulating platelet-derived antimicrobial peptides. There is significant heterogeneity in the catalog of virulence factors produced by different *S. aureus* clinical isolates [57,58,59,60], the regulators mediating virulence factor expression [61,62,63,64], and susceptibility to antimicrobial peptides [65,66,67,68]. This section discusses the key genetic and phenotypic characteristics of *S. aureus* that have been associated with persistent SAB.

### 3.1. Accessory Gene Regulator Dysfunction 

Virulence factor production is tightly controlled by a series of regulatory mechanisms including several two-component systems and SarA-family regulators [69]. One of the most well-characterized global regulators of virulence factor production is the two-component quorum-sensing accessory gene regulator (*agr*) system of *S. aureus* [70]. The *agr* system is a quorum-sensing system that mediates expression of exotoxins and exoenzymes [69]. The essentiality of *agr* to virulence in *S. aureus* infection depends on the type of infection [70]. Murine skin and soft tissue models have shown that *agr* deletion mutations are severely attenuated. However, *agr*-null *S. aureus* strains are frequently isolated from the bloodstream of human subjects with SAB [16,61,62,63,71,72,73]. Several groups have shown that specific *agr* genotypes are associated with persistent MRSAB [16,74,75]. Fowler et al. discovered that isolates from patients with persistent MRSAB were predominantly (≈85%) of similar *agr* genotypes and lacked *agr* activity, as measured by δ-lysin production. The same study also noted that isolates from patients with persistent MRSAB were less susceptible to killing by thrombin-induced platelet microbicidal protein, an antimicrobial peptide produced by host platelets. Another study by Park et al. examined the *agr* genotype in MRSAB patients without retained foci of infection (e.g., prosthetic joint, intravenous catheter) [74]. They found that persistent MRSAB isolates more frequently possessed *agr* dysfunction compared to those from patients with resolving bacteremia (94% vs. 75%, *p* = 0.03). A third investigation by Kang et al. limited their investigation to 152 patients with persistent MRSAB and asked if infections due to isolates with *agr* dysfunction had worse clinical outcomes compared to *agr* positive strains [75]. They found significantly higher rates of in-hospital mortality in patients with persistent MRSAB if the bloodstream isolate had a dysfunctional *agr* system (68% vs. 49%, *p* = 0.029). The mechanism for the reciprocal relationship between *agr* activity and persistence remains unclear but is likely multifactorial. First, the reduction in cytotoxic leukocidin production in *agr*-null isolates may lead to decreased host-cell toxicity and increased bacterial survival [75]. Second, the *agr* operon also repressed adhesins such as *fnbA*, which are required for adhesion and invasion of endothelial cells. The lack of a functional *agr* would result in upregulated adhesins and potentially enhanced intracellular invasion, where it would be shielded from the effects of numerous antibiotics including vancomycin. Third, multiple studies have linked *agr* dysfunction with glycopeptide intermediate-resistance or vancomycin tolerance (discussed further in Section 3.4 The mechanism of increased antibiotic tolerance is thought to be due to altered autolysin activity, blunting the bactericidal effect of vancomycin [61,74]. These studies provide some compelling evidence that *agr* dysfunction can be a driver of persistent SAB.

### 3.2. Variability in Virulence Factor Production

Despite several decades of mechanistic studies examining *S. aureus* virulence factor function and regulation, the field has been unable to pinpoint which specific virulence factors are responsible for microbial survival in bloodstream infections. It appears that no single virulence factor can dictate the pathophysiology, which points towards combinations that are likely expressed in different infectious niches. Few studies have examined virulence factor expression to specifically differentiate persistent MRSAB from resolving MRSAB isolates. Xiong et al. performed an in vitro analysis on isolates from patients with persistent MRSAB and resolving MRSAB to determine phenotypic characteristics that may distinguish the two isolates [76]. They found that isolates from persistent MRSAB patients differed in several characteristics. First, the persistent MRSAB isolates were more resistant to killing by hNP-1, an antimicrobial peptide produced by neutrophils. Second, they discovered that persistent MRSAB isolates were more adept at binding to fibrinogen and fibronectin, which are thought to act as the anchors allowing *S. aureus* to establish endovascular infection. Third, multiplex genotyping identified the genes *cna, sdrD*, and *sfrE* more frequently in persistent MRSAB isolates compared to resolving MRSAB isolates. However, another larger study using the same definition of persistent MRSAB (cultures positive >7 days) was unable to find differences in the presence of virulence factor genes (including *sdrD*) or *agr* dysfunction [22]. Similarly, Seidl et al. did not note any differences in fibronectin binding between persistent versus resolving MRSAB isolates [77]. These inconsistencies between studies may highlight epidemiological differences between SAB isolates from different geographic centers.

### 3.3. Phenotypic Variability of SAB Isolates

While genotypic analysis has been extremely informative in differentiating persistent MRSAB from resolving MRSAB isolates, often the downstream effects on function are a result of multiple interacting processes. Following on from Xiong et al.’s work discussed in Section 3.2, Seidl et al. performed several in vitro studies to distinguish functional differences between isolates from patients with persistent MRSAB vs. resolving MRSAB [77]. They again confirmed that persistent MRSAB isolates exhibited significantly less killing by the neutrophil-derived AMP hNP-1 (*p* = 0.02) and platelet-derived thrombin-induced platelet microbicidal proteins (tPMPs, *p* = <0.001). Other findings from the study noted no significant difference in overall biofilm biomass produced, but they did report biofilms from persistent MRSAB isolates contained a lower carbohydrate content (58.4% vs. 30.6%; *p* = 0.04). It is thought that platelet-derived antimicrobial peptides, such as tPMPs, play a key role in assisting clearance of *S. aureus* in the bloodstream, particularly around areas of endothelial damage that are thought to serve as an anchor in the establishment of an endovascular infection [78]. *S. aureus* isolates exhibiting decreased killing by tPMPs in-vitro show increased virulence in an in vivo rabbit endocarditis model [66,79]. Furthermore, *S. aureus* bloodstream isolates from patients with confirmed endovascular infections were less susceptible than bacteremia strains without an endovascular source [67,68]. It is reassuring to see the clinical relevance of the in vitro studies by establishing the relationship between decreased tPMPs killing and persistent MRSAB [16,76]. The relationship between decreased hNP-1 killing and persistence is less well established but could be a result of increased survival inside neutrophils after phagocytosis [76,77]. 

### 3.4. Antibiotic Tolerance 

Antibiotic resistance is the inherited ability of bacteria to grow in the presence of elevated concentrations of antibiotics and is quantified by measuring the minimum inhibitory concentration (MIC). Antibiotic tolerance refers to the ability of a population of bacterial cells to survive in the presence of lethal concentrations of bactericidal antibiotics without a change in the MIC [80]. Resistance generally involves a specific mechanism, such as modification of the target, efflux pumps, or deactivation of the antibiotic, whereas the mechanisms of antibiotic tolerance are more general and are commonly associated with slower growth and decreased metabolic activity. The absence of MIC alteration and the wide variability in the pathways that lead to tolerance means the phenotype is challenging to detect. There is currently no standardized testing protocol allowing for detection of antibiotic tolerance in the clinical microbiology laboratory. Additionally, tolerance is highly dependent on the environment, making it difficult to measure under ex vivo conditions. Studies have shown a proportion of *S. aureus* can survive phagocytosis by host immune cells and persist in the intracellular space [81]. Due the poor intracellular permeability of antibiotics such as vancomycin and daptomycin, these intracellular bacteria are shielded from the effects of serum antibiotics [82]. Recent work by Rowe et al. discovered that host immune cells can also induce antibiotic tolerance in *S. aureus* by ROS-mediated inactivation of key tricarboxylic acid cycle (TCA) enzymes [83,84]. Another mechanism of host-induced tolerance was identified by Ledger et al., who report that human serum can induce daptomycin tolerance through LL-37-mediated activation of the GraRS two-component system and membrane lipid remodeling [85]. These studies emphasize the diversity in the mechanisms of antibiotic tolerance and underline the difficulty of detecting these phenotypes once the bacteria is removed from the host environment. The most common method for determining antibiotic tolerance is by performing a time-kill curve, which looks at the rate of antibiotic killing of a pathogen by an antibiotic over time [86], which is laborious and not feasible in a busy clinical microbiology laboratory. The devastating consequences of antibiotic resistance are ubiquitously acknowledged through the scientific community, although the clinical impact of antibiotic tolerance is less well understood. In addition, there is no standardized definition of antibiotic tolerance, although some groups have agreed that a minimum bactericidal concentration (MBC) to MIC ratio of >32 is consistent with tolerant bacteria [87,88,89,90]. A key study by Levin-Reisman revealed that antibiotic tolerance acts as a precursor to antibiotic resistance [91]. The mechanism proposes that decreased antibiotic killing in antibiotic-tolerant cells results in an increase in the pool of viable cells available to acquire mutations that confer resistance. Further studies are needed to explore if this phenomenon can be extrapolated beyond ampicillin tolerance and resistance in *Escherichia coli*. While the clinical relevance of this finding will require further experiments, it provides further evidence that tolerance may be an unappreciated pathway to treatment failure [91].

Glycopeptide tolerance has been frequently observed in *S. aureus*, with a prevalence of up to 43% in MRSA isolates [87,92]. While it is suspected that antibiotic tolerance is a contributor to refractory and relapsing infections, there are few studies that have directly addressed this question. Given the definition of decreased antibiotic killing in antibiotic tolerance, one could hypothesize that antibiotic tolerance may play a role in persistent bacteremia. Britt et al. performed a retrospective cohort study of 225 patients with SAB comparing frequency of clinical failure (30 day all-cause mortality, persistent signs and symptoms of bacteremia, recurrent bacteremia within 30 days, and blood culture positive >5 days) between isolates with and without vancomycin tolerance [88]. In their study, 26.7% of the isolates exhibited vancomycin tolerance, which was associated with clinical failure in unadjusted (68.3% vs. 40.6%) and multivariable analysis (adjusted risk ratio, 1.74; 95% CI, 1.35–2.24; *p* < 0.001). The average bacteremia duration did not significantly vary between the two groups, nor did the proportion with blood cultures positive for >3 days (48.2% in vancomycin-tolerant (VT) vs. 38.4% in non-VT). Another smaller study of 163 patients with MRSAB from St. Louis, USA, noted just 4.3% of isolates were vancomycin-tolerant with no statistically significant effect on clinical outcomes. Finally, a study by Moise et al. noted increased duration of bacteremia (median time to clearance 6.5 days vs. >10.5 days, *p* = 0.025) when MRSA isolates were stratified by tolerance (≤2.5 log10 decrease in colony-forming units/mL over 24 h of vancomycin treatment) [93]. Larger studies are needed to determine the clinical impact of antibiotic tolerance in persistent MRSAB.

The mechanisms of antibiotic tolerance are incompletely understood, especially in *S. aureus*. To identify if antibiotic tolerance evolves within patients, Elgrail et al. performed WGS on 206 MRSA isolates from 20 patients with persistent MRSAB [94]. Their results showed that MRSA can evolve antibiotic tolerance within the host due to mutations in the TCA cycle (*odhA* and *citZ*) and stringent response (*relA*). Interestingly, these mutants were transient and were not present in subsequent positive blood cultures, suggesting there is phenotypic heterogeneity and a fitness cost to tolerance, which has been described in other pathogens [95]. 

### 3.5. Reduced Vancomycin Susceptibility and Heterogenous Vancomycin-Intermediate S. aureus

Vancomycin is the oldest and most frequently used drug in our arsenal against MRSA [96]. Despite being used for almost 65 years, vancomycin resistance (MIC ≥ 16 μg/mL) is extraordinarily uncommon, with just 52 incidents of vancomycin-resistant *S. aureus* (VRSA) reported worldwide in the past two decades [97]. Vancomycin-intermediate *S. aureus* (VISA) is defined by a vancomycin MIC between 4 and 8 μg/mL and is more frequent with an estimated prevalence of between 0.3 and 18% depending on the geographic area [98]. In theory, vancomycin is an appropriate treatment for MRSAB isolates with vancomycin MIC between 1 and 2 μg/mL. There has been a long-standing debate questioning whether MRSA with elevated vancomycin MIC (>1.5 μg/mL) is associated with worse clinical outcomes or not. The majority of data, including two systematic reviews and meta-analyses, indicates that MRSAB due to isolates with high vancomycin MIC (>1.5 μg/mL) is associated with increased mortality compared to MRSAB due to isolates with low-vancomycin MIC (<1.5 μg/mL) [93,99,100]. This finding is not necessarily related to failure of vancomycin, as an elegant study by Holmes et al. also found worse clinical outcomes in MSSA bacteremia isolates with elevated vancomycin MIC, despite treatment with flucloxacillin and not vancomycin [101]. This finding is consistent with the Infectious Disease Society of America (IDSA) recommendations to base treatment decisions in patients infected with MRSA isolates with vancomycin MIC of 2 μg/mL upon clinical conditions [91]. The majority of studies examining the risk of elevated vancomycin MIC with clinical outcomes used composite outcomes for treatment failure, often including (but not always specifying) persistent bacteremia [100]. When the systematic review and meta-analysis by van Hal et al. limited their analysis exclusively to studies that examined persistent MRSAB, the OR was 2.44 but was not significant (95% CI, 0.72–8.24) [100]. Some individual studies did show an association, such as a retrospective cohort of 222 MRSAB patients by Neuner et al. that identified a significantly higher rate of persistent MRSAB when vancomycin MIC was 2 μg/mL compared to <2 μg/mL (16% vs. 5 %, *p* = 0.012) [102]. Another smaller study by Yoon et al. also found vancomycin MIC of 2 μg/mL is an independent predictor of persistent MRSAB (OR 6.34; 95% CI, 1.21–33.09) [65]. Another newer study by Adani et al. of 166 patients from an institution with blinded vancomycin MIC showed no significant difference in persistent bacteremia rates between isolates with MIC < 2 μg/mL vs. 2 μg/mL (16.5% vs. 17.3%, *p* = 0.884) [103]. 

Heterogenous VISA (hVISA) is another microbiologic phenomenon that could contribute to decreased vancomycin efficacy [104]. The first reported case of hVISA was in 1996 from a patient in Japan with MRSA pneumonia that did not respond to vancomycin [105]. Despite susceptibility testing showing vancomycin MIC of 4 μg/mL, a subpopulation was discovered with MICs ranging from 5 to 9 μg/mL. An isolate with vancomycin MIC in the susceptible range (≤2 μg/mL) with a subpopulation with vancomycin MIC in the intermediate range (4–8 μg/mL) has become diagnostic of hVISA [106]. Similar to the challenges of identifying antibiotic tolerance, the detection of hVISA is laborious and utilizes the population analysis profile (PAP) area under the curve (AUC) technique, which is not feasible in the clinical microbiology lab on a routine basis [104]. It was previously thought that hVISA is a precursor to VISA as selection pressure during treatment with vancomycin generates outgrowth of the VISA subpopulation [107,108], although more recent data from in vitro evolutionary experiments suggests that may not be correct [109]. Whether hVISA in MRSAB results in increased vancomycin failure and persistent MRSAB remains debated. Some studies report worse clinical outcomes [110,111,112,113,114,115,116] and increased risk of persistent MRSAB [110,112,113,114], with others, including one systematic review and meta-analysis, showing no significant difference in mortality or persistent MRSAB [104,117,118,119,120,121]. Overall, the mixed data suggest that hVISA may play a role in persistent MRSAB. However, the lack of strong evidence does not necessarily justify deviating from vancomycin in routine hVISA MRSAB cases.

In summary, there is unlikely to be a single pathogen component that is individually responsible for persistence in MRSAB. The inability of the host to clear the bloodstream is likely a result of complex interplay between the bacteria, the host immune system, and the circulating antibiotic (Figure 3). Understanding characteristics of *S. aureus* increasing the probability of persistent bacteremia opens the door to novel diagnostics, which could allow for a more aggressive antibiotic strategy up-front, potentially improving patient outcomes.

## 4. Treatment of Persistent MRSAB

Limited high-quality evidence exists for the most effective treatment of MRSAB in general, and even less for the treatment of persistent MRSAB in particular [122]. No randomized controlled trials to date have addressed this specific question, leaving an unmet need for medical practice. However, until high-quality evidence is available, the available literature provides suggestions for best practice regarding the treatment of persistent MRSAB.

The management of MRSAB consists of three important pillars: source control, antibiotic treatment, and follow-up blood cultures. For evaluation of metastatic infection sites as targets for source control, the transesophageal echocardiogram is the most evidence based [123,124]. For positron emission tomography/computed tomography (PET-CT), there is evidence for impacting management and for reducing mortality in patients with SAB [125,126], although this latter finding may have been confounded by the introduction of immortal time bias related to including patients dying before undergoing PET-CT. Thorough clinical assessment by a trained infectious diseases consultant has been proven to be beneficial in the management of MRSAB [127]. In the case of positive follow-up blood cultures and thus persistent bacteremia despite adequate treatment, potential targets for source control must be reevaluated, and subsequently also the antibiotic therapy. This is particularly true now, as the specific antibiotic treatment options have evolved over time. 

### 4.1. The Past

For decades, vancomycin monotherapy was the only recommended antibiotic treatment for MRSAB. This was primarily due to the lack of other options for monotherapy. There has been a multiplicity of attempts to craft an effective combination antibiotic therapy for SAB. Adjunctive gentamicin appeared to be an attractive option according to in vitro data, but was associated with increased nephrotoxicity without any clinical benefit [128]. Alternatives for vancomycin, such as trimethoprim-sulfamethoxazole, did not achieve non-inferiority for the treatment of MRSAB [18,129]. For many years, the addition of rifampin was thought to improve outcomes, but the ARREST trial has ruled out that hypothesis: outcomes in both MSSA and MRSAB did not improve with adjunctive rifampin [130].

Historically, there were few options for treatment of persistent MRSAB. When confronted with persistent MRSAB > 7 days after vancomycin initiation and a MIC of 2 μg/mL, almost three-quarters of surveyed American ID consultants in 2005 would continue vancomycin and add another drug, usually rifampin or gentamicin. Less than 20% would switch to another agent [131]. Rather than clinical inertia, this approach was likely a consequence of the paucity of agents with proven efficacy for SAB. This changed in 2006, when daptomycin was proven to be non-inferior to vancomycin in the treatment of MRSAB [19].

### 4.2. The Present

Following the non-inferiority trial in 2006, the U.S. guideline included daptomycin as first-choice therapy, comparable to vancomycin, for MRSAB in 2011 [10,19]. Although daptomycin monotherapy was shown to be non-inferior to vancomycin for treatment of MRSAB, the possibility of treatment-emergent resistance and treatment failure has become apparent over time [132,133]. Therefore, it is often recommended to add a second antibiotic agent to daptomycin (e.g., trimethoprim-sulfamethoxazole) with the goal of preventing daptomycin resistance from emerging, especially if source control is not achieved [10]. In Europe and the UK, the only first-choice agent in the guidelines remains vancomycin [134,135]. However, when the MIC is 2 μg/mL or higher, vancomycin is believed to be less effective, and alternative treatment options should be considered. 

Multiple mono- or combination therapy options for the treatment of MRSAB have been studied in the last decade. One promising concept was the combination of vancomycin or daptomycin with an anti-staphylococcal beta-lactams (ASBLs) such as nafcillin or flucloxacillin. This clinical approach was based on exciting in vitro data demonstrating the synergy with both vancomycin and daptomycin when an ASBL was added. The CAMERA2 trial addressed this question by randomizing MRSAB patients to receive either standard therapy (daptomycin or vancomycin) or standard therapy with the addition of an ASBL. While the proportion of patients with persistent *S. aureus* bacteremia at day five was significantly lower in the combination therapy group, all-cause mortality was not significantly different and combination therapy was associated with a significantly increased rate of acute kidney injury [136]. However, whether this is true for all beta-lactams and for all patient categories has not yet been clarified [137]. The DASH trial, which enrolled only MSSA bacteremia patients, demonstrated that the addition of daptomycin to anti-staphylococcal beta-lactam did not reduce the duration of bacteremia, 90-day mortality, or rate of recurrence [138]. 

Ceftaroline is a fifth-generation cephalosporin with robust activity against MRSA due to its unique ability to bind with high affinity to PBP-2a [139]. It is FDA approved for the treatment of community-acquired pneumonia and acute bacterial skin and skin structure infections (including those with concurrent bacteremia) but is frequently used off-label, either alone or in combination with another antibiotic, as a treatment for MRSAB. The combination of daptomycin and ceftaroline, especially when initiated early in the disease course, is possibly associated with reduced in-hospital mortality compared to monotherapy with vancomycin or daptomycin [140,141,142]. Although we are lacking high-quality data to support such an approach, ceftaroline is commonly used in clinical practice in combination with vancomycin or daptomycin to treat persistent MRSAB [143,144]. There are several observational studies showing expedited bacterial clearance when deployed as a salvage therapy in refractory MRSAB, but the effect on mortality remains unclear [145,146,147,148,149]. Fortunately, a large, well-designed Phase 3 randomized clinical trial that tested ceftobiprole, another cephalosporin with efficacy against MRSA, has recently completed enrollment and reported positive topline results (discussed later). 

The emergence of possible alternatives for the treatment of MRSAB has an effect on the decisions that physicians make in clinical practice. In contrast to the situation in 2005, a second survey in 2017 showed that less than 20% of the surveyed American ID consultants would continue vancomycin and simply add another agent in case of persistent MRSAB on day 6. Instead, more than half of them would switch to another agent (either a single agent or daptomycin with a second agent) [150]. 

Although there is much (clinically unsubstantiated) debate about the most appropriate therapeutic modification in patients with persistent MRSAB, the single most important management component of these patients remains adequate source control. In the suggested management algorithm for MRSAB by Holland et al., a single positive follow-up blood culture represents a “worry point”, prompting reevaluation of potential sites of metastatic infection [21]. If blood cultures continue to be positive at the 3–5-day point despite appropriate antibiotic therapy, Holland et al. presume the patient has experienced monotherapy failure and recommend the addition of ceftaroline to vancomycin or a change of therapy to daptomycin plus a second antibiotic. The recommendation to add a second antibiotic to daptomycin or vancomycin, while unproven, is primarily to thwart the development of treatment-emergent daptomycin resistance rather than to improve efficacy based upon data using simulated vegetations [151]. 

### 4.3. The Future

There are a handful of clinical trials investigating future therapeutics for the treatment of MRSAB. Ceftobiprole is another fifth-generation cephalosporin currently under investigation with activity against MRSA [152,153]. Its safety and efficacy were recently evaluated in a landmark clinical trial. The ERADICATE trial is the largest clinical trial to evaluate a new antibiotic for complicated SAB and the first double-blind, placebo-controlled Phase 3 ever conducted for that indication [154]. Results were presented at IDWeek2022. Topline data from the ERADICATE trial indicate that ceftobiprole met its primary efficacy endpoint without significant obvious toxicity concerns. 

Dalbavancin is approved for use in *S. aureus* bacterial skin infections, with the great advantage of having a uniquely long half-life [155]. A potential role of dalbavancin in endovascular infections has not yet been established [156]. The superiority of dalbavancin compared to standard parenteral antibiotic therapy for the completion of treatment is currently being studied in patients with complicated SAB in a phase 2b randomized clinical trial (DOTS trial) [157]. A potential role for dalbavancin in persistent bacteremia naturally warrants more follow-up research. 

Driven by the lack of major breakthroughs in antibiotic treatment to improve clinical outcomes in SAB, new nonantibiotic antimicrobial modalities are an increasing subject of research. Exebacase, an anti-staphylococcal lysin, as an addition to standard-of-care antibiotics, led to a higher clinical response rate in patients with MRSAB in a proof-of-concept study [158]. A subsequent randomized trial addressing the superiority of exebacase in addition to standard-of-care antibiotics in both MSSA and MRSAB (DISRUPT trial) was terminated early for futility, following interim efficacy analysis [159]. A second anti-staphylococcal lysin, LSVT-1701, showed reduced bacterial bioburden in MRSA animal studies and demonstrated a good safety profile in a Phase I study in healthy human subjects [160]. In June 2022, further development of this asset was terminated by Roivant Sciences. Furthermore, bacteriophage therapy as an adjunctive intravenous therapy for SAB patients is currently being investigated. It was shown to be well tolerated in a group of 13 patients with severe *S. aureus* infections, including endocarditis and septic shock [161]. The diSArm trial is a phase 1b/2a randomized trial on the efficacy and safety of adjunctive bacteriophage therapy in SAB patients, which is estimated to be completed at the end of 2023 [162].

In conclusion, the unfavorable safety profiles of many combinations of antibiotics have prevented them from replacing vancomycin as the most frequently used antibiotic treatment in MRSAB. High-dose daptomycin (with a second antibiotic agent to prevent treatment-emergent resistance) and the addition of ceftaroline are currently the best practice in persistent MRSAB. Future treatment options may include dalbavancin, ceftobiprole, and novel non-antibiotic agents such as bacteriophages.

## 5. Conclusions

Persistent MRSAB is a devastating and complex disease. Understanding the interaction between host and pathogen is crucial to the challenge of improving patient outcomes. Given the lack of major breakthroughs in patient outcomes in the last decades, there seems to be a need for novel diagnostics and treatment options. Trials on genetics, biomarkers, and novel non-antibiotic agents in persistent MRSAB should be encouraged, as well as the implementation in daily practice of those that were successful. Meanwhile, it is promising that antibiotic agents such as dalbavancin [157] and ceftobiprole [154] are being studied in randomized clinical trials for SAB. These new high-quality studies represent an important step towards better understanding and ultimately improving clinical outcomes in patients with SAB.

## Figures and Tables

**Figure 1 antibiotics-12-00455-f001:**
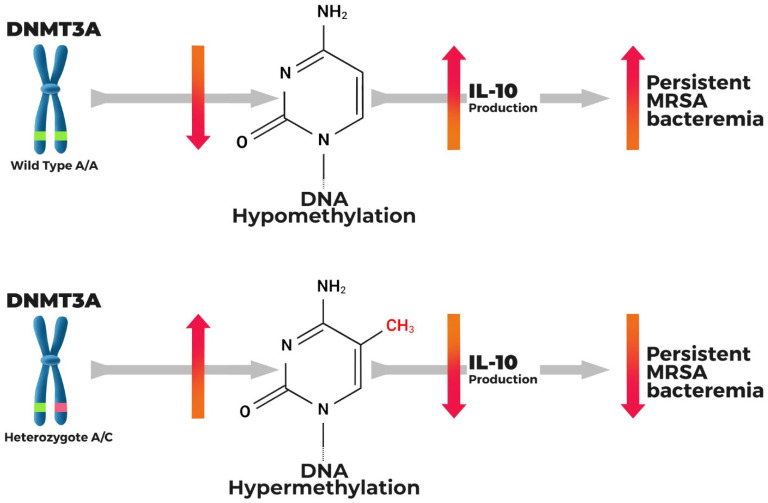
Proposed relationship of DNMT3A polymorphisms and increased risk of persistent MRSAB. Created using Biorender.com.

**Figure 2 antibiotics-12-00455-f002:**
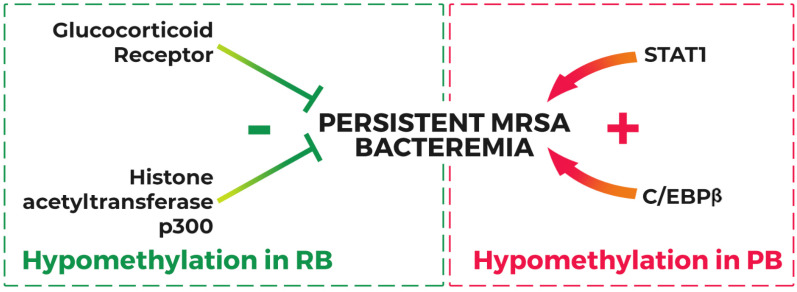
Schematic showing genes with hypomethylation in patients with persistent MRSA bacteremia (PB) and resolving MRSA bacteremia (RB).

**Figure 3 antibiotics-12-00455-f003:**
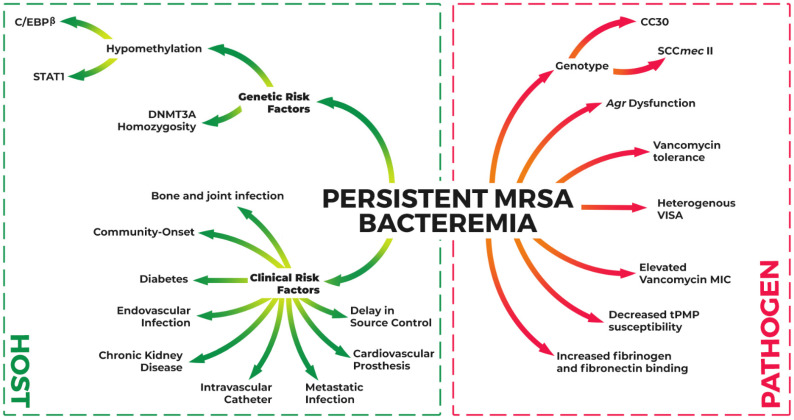
Summary of host and pathogen factors contributing to persistent MRSAB.

**Table 1 antibiotics-12-00455-t001:** Clinical risk factors for persistent SAB.

Study	Year	MSSA or MRSA	Definition of Persistent Bacteremia	Clinical Risk Factors Identified
Khatib et al. [26]	2006	MSSA and MRSA	3 days	Intravascular catheter (RR, 1.27; 95% CI 1.03–1.54) Cardiovascular prosthesis (RR, 1.24; 95% CI 0.97–1.59) Metastatic infection (RR, 1.16; 95% CI 1.05–1.28)
Hawkins et al. [25]	2007	MSSA and MRSA	7 days	Chronic renal failure (OR, 2.08; 95% CI 1.09–3.96) >2 sites of infection (OR, 3.31; 95% CI 1.17–9.38) Infective endocarditis (OR, 10.3; 95% CI 2.98–35.64) Presence of intravascular catheter or foreign device (OR, 2.37; 95% CI 1.11–3.96)
Khatib et al. [23]	2009	MSSA and MRSA	7 days	Metastatic infection (OR, 5.6; 95% CI 3.00–10.47) Vancomycin treatment (OR, 4.17; 95% CI 2.14–8.11) Endovascular source (OR. 3.35; 95% CI 1.92–5.85) Diabetes (OR, 2.14; 95% CI 1.26–3.64)
Ganga et al. [28]	2009	MRSA and MSSA	7 days	Metastatic infection (OR, 11.35; 95% CI 4.24–31.43 Diabetes (OR, 3.64; 95% CI 1.45–9.155) Prosthetic device (OR, 3.22; 95% CI 1.30–8.00)
Yoon et al. [24]	2010	MRSA	7 days	Retention of infected medical device (OR, 10.35; 95% CI 1.03–104.55) Infection of at least two metastatic sites (OR, 10.24; 95% CI 1.72–61.01)
Chong et al. [22]	2013	MSSA and MRSA	7 days	Community-onset bacteremia (OR, 2.91; 95% CI, 1.24–6.87) Bone and joint infection (OR, 5.26; 95% CI, 1.45–19.03) Central-venous-catheter-related infection (OR, 3.36; 95% CI, 1.47–7.65) Metastatic infection (OR, 36.22; 95% CI, 12.71–103.23) Delay in removal of eradicable foci >3 days (OR, 2.18; 95% CI, 1.05–4.55)

Abbreviations: MSSA, methicillin-susceptible *Staphylococcus* aureus; MRSA, methicillin-resistant *Staphylococcus aureus;* RR, risk ratio; CI, confidence interval; OR, odds ratio.

## Data Availability

No new data were created or analyzed in this study. Data sharing is not applicable to this article.

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
