# Peer review of "Persistent Methicillin-Resistant Staphylococcus aureus Bacteremia: Host, Pathogen, and Treatment"

_antibiotics, 2023, doi:10.3390/antibiotics12030455_

Round 1

Reviewer 1 Report

The article by Parsons and colleagues reviews all important topics related to persistent MRSA bacteremia; biomarkers, risk factors, features related to bacterial strains or to host genetics …. It was a real pleasure to read this well-written review containing an important amount of information allowing to evaluate the importance of the field.

I have only very minor comments for the authors:

-          P1 : MRSA/MSSA bacteremia :

MRSA BSI has only a slightly higher adjusted mortality compared to MSSA BSI. High quality studies in the field suggest a limited OR or RR increase of deaths around 1.3-1.5.

P9: Antibiotic tolerance

“ATB tolerance acts as a precursor of resistance should be modulated to specific antibiotics only.

Based on the increasing number of bacterial genome sequences available in public database, are the authors aware of any comparative genomic study on persistent/resolving strains?

Table 1 and following pages 3, 4 need editing

Author Response

  1. P1 : MRSA/MSSA bacteremia :

MRSA BSI has only a slightly higher adjusted mortality compared to MSSA BSI. High quality studies in the field suggest a limited OR or RR increase of deaths around 1.3-1.5.

We thank the reviewer for the comment. We have added the following statement to page 1: “Although most studies have shown higher mortality rates, MRSA bacteremia (MRSAB) has only a slightly higher adjusted mortality compared to methicillin-susceptible SAB More recent high-quality studies in the field suggest a limited odds ratio (OR) or relative risk (RR )increase in death around 1.3 – 1.8” and added the following citation (Bai et al, Clin Microbiol Infect, 2022) to support these data.

  1. P9: Antibiotic tolerance

“ATB tolerance acts as a precursor of resistance should be modulated to specific antibiotics only.

We thank the reviewer for the comment. We added an additional statement on P9 “Further studies are needed to explore if this phenomenon can be extrapolated beyond ampicillin tolerance and resistance in Escherichia coli.”

  1. Based on the increasing number of bacterial genome sequences available in public database, are the authors aware of any comparative genomic study on persistent/resolving strains?

We thank the reviewer for the thoughtful comment. Our group has performed such a study comparing the bacterial genomic signature of resolving versus persistent MRSA isolates, which was presented as a poster at ID week in 2019 (https://doi.org/10.1093/ofid/ofz360.705). We did not see any bacterial genetic signature associated with either PB or RB. To our knowledge, this is the only study of its kind and remains unpublished.

  1. Table 1 and following pages 3, 4 need editing.

Thank you for the comment. We have corrected the formatting errors.

Reviewer 2 Report

Great novel review that focusses on the clinical aspects of persistent  Staphylcoccal infections. The only comments I  have to make this review more comprehensive is the notion that  MRSA can also reside intracellular and this   would limit the therapeutic efficacy of Vancomycin. Lehar et al 2015 and Surewaard et al 2016.

minor comment:

line 121-123 is a different font 

Author Response

Reviewer 2:

  1. Great novel review that focusses on the clinical aspects of persistent  Staphylcoccal infections. The only comments I  have to make this review more comprehensive is the notion that  MRSA can also reside intracellular and this   would limit the therapeutic efficacy of Vancomycin. Lehar et al 2015 and Surewaard et al 2016.

We thank the reviewer for the comment. We have added the Lehar and Surewaard citation and the following statement to page 13: “Studies have shown a proportion of S. aureus can survive phagocytosis by host immune cells and persist in the intracellular space. Due the poor intracellular permeability of antibiotics such as vancomycin and daptomycin, these intracellular bacteria are shielded from the effects of serum antibiotics”

  1. line 121-123 is a different font

Thank you, the font standardized.